# Application of Ultrafiltration and Ion Exchange Separation Technology for Lysozyme Separation and Extraction

**Shanshan Chen, Yaqing Tan, Yaqing Zhu, Liqin Sun, Jian Lin * and Hailing Zhang ***

College of Life Sciences, Yantai University, Yantai 264005, China; 18865557198@163.com (S.C.); tanyaqing371122@163.com (Y.T.); zhuyaqing0524@163.com (Y.Z.); sliqin2005@163.com (L.S.)
* Correspondence: linjian3384@163.com (J.L.); hailing1203@hotmail.com (H.Z.); Tel.: +86-0535-6902638 (H.Z.)

**Abstract:** In this study, the fermentation broth of the recombinant *Pichia pastoris* strain ncy-2 was studied. After pretreatment, separation, and purification, lysozyme was optimized using biofilm and ion exchange separation. Finally, lysozyme dry enzyme powder was prepared by concentrating and vacuum drying. The removal rate of bacterial cells was 99.99% when the fermentation broth was centrifuged at low temperature. The optimum conditions were: transmembrane pressure of 0.20 MPa, pH 6.5, 96.6% yield of lysozyme, enzyme activity of 2612.1 u/mg, which was 1.78 times higher than that of the original enzyme; D152 resin was used for adsorption and elution. Process conditions were optimized: the volume ratio of resin to liquid was 15%; the adsorption time was 4 h; the concentration of NaCl was 1.0 mol/L; the recovery rate of lysozyme activity was 95.67%; the enzyme activity was 3879.6 u/mL; and the purification multiple was 0.5, 3.1 times of the original enzyme activity. The enzyme activity of lysozyme dry enzyme powder was 12,573.6 u/mg, which had an inhibitory effect on microsphere lysozyme. Its enzymatic properties were almost the same as those of natural lysozyme, which demonstrated good application prospects and production potential.

**Keywords:** lysozyme; biofilm; ion exchange resin; separation; purification

## 1. Introduction

Lysozyme is a kind of natural lyase, which can specifically hydrolyze the peptidoglycan structure of the cell wall of many microorganisms, especially Gram-positive bacteria. Its antibacterial protection mechanism is significant. It has been a long development process since Nicolle first isolated the dissolving factor from *Bacillus subtilis*, and the World Health Organization (WHO) and many countries and regions determined that lysozyme can be used as a non-toxic and safe additive. Nowadays, lysozyme is widely used in medicine [1], food, scientific research, and other fields [2–4], especially in a variety of industries. Therefore, the production and purification of lysozyme has become very important. Lysozyme is an alkaline protein with stable chemical properties. It can maintain its original structure and activity under a wide temperature and pH range. It has a high isoelectric point and is mostly positively charged. It is a high molecular weight compound with a variety of dissociable multivalent amphoteric electrolytes. It has different amounts of positive or negative charges at different pH values. In acidic and near neutral environments, there are many positive charges. Most other proteins are acidic proteins. When they coexist with other proteins, the interaction between molecules is easy to combine into a certain macromolecule, so it is difficult to separate lysozyme directly.

Lysozyme, because of its non-specificity, can be used as a cellular immune protein in various organisms, such as birds, mammals, and bacteria. Among them, the content inegg white is particularly rich. The traditional process usually adopts the combination of ultrafiltration and chromatography to remove the force of enzyme molecules and other proteins, so as to achieve the purpose of separating lysozyme. The common way to obtain lysozyme is to use egg white as raw material. However, the process of extracting lysozyme

from egg white is limited by the source of egg white. Because egg white has only about 3.5% lysozyme, extraction has high production cost, it is a complicated operation, and it has very limited output and low profit, which is not conducive to amplification, and it is difficult to realize large-scale industrial production. Therefore, it is necessary to produce substitutes and reduce production costs to solve the problem of reducing enzyme production. Microbial fermentation with genetically engineered bacteria is an effective way to solve the above problems. However, it is an important challenge to separate lysozyme from the fermentation broth, because the metabolism of microorganisms involves multiple integrated processes, so that in addition to lysozyme, there are many complex components in the fermentation broth: water, residual substrates, by-products, and macromolecules (such as protein and polysaccharide). The first step in the treatment of fermentation broth is to remove various insoluble impurities such as microbial cells and residual substrates, which can be solved by centrifugal filtration. Then, it is necessary to remove the impurity protein and other macromolecular substances in the clear liquid as much as possible. Ultrafiltration can be used for further treatment. Ultrafiltration membrane technology can push water and small molecular substances through the membrane and release them into the permeate according to the molecular weight of the target product and the pressure difference between the two sides of the membrane; lysozyme is intercepted during this process. It is an excellent fermentation liquid purification technology. Important aspects include obtaining clearer permeate; less energy demand; recyclable material; simple and efficient operation; improved purity; and assuredlysozyme activity [5,6]. However, the viscous substances in the feed liquid are easy to adsorb, block the membrane pores, and form a filter cake layer, resulting in concentration polarization on the surface of the ultrafiltration membrane, increasing the resistance and affecting the transmittance, rejection, and membrane flux. Therefore, adjusting the ionic strength in the feed liquid, weakening the force between molecules, selecting the appropriate ultrafiltration membrane pressure, and improving the membrane flux are of great significance to improving the purity of the feed liquid. Lysozyme cannot be completely separated from other impurities in the feed solution by using biofilm, and it needs further refining and purification. Ion exchange technology can complete the refining and purification process, select the appropriate resin as the filler, and use the difference of binding force between lysozyme and exchange groups in the resin to complete the purification of lysozyme in the process of adsorption, binding, and elution. There are many factors affecting the adsorption–desorption process, such as resin type, time, eluent concentration and dosage, etc.,and optimizing the operating conditions of ion exchange is particularly important for the commercial production of lysozyme [7]. The whole process is easy to operate, has low cost, has no need to add any chemical reagent, and is safe and efficient. In particular, ultrafiltration technology has mild conditions; does not cause changes in temperature and pH; can prevent denaturation, inactivation, and autolysis of lysozyme molecules; ensures the activity of lysozyme to the greatest extent; and provides good contact conditions for ion exchange. The macroporous resin has many and large pores, large surface area, many active centers, fast diffusion speed, short distance, high efficiency, short processing time, easy adsorption and exchange, strong pollution resistance, and a stable structure, and is renewable and recyclable [8]. The purpose of this study is to combine ultrafiltration membrane and ion exchange technology, improve the extraction process, seek the best process parameters, ensure the enzyme activity to the greatest extent, extract lysozyme step by step from the fermentation broth, purify and refine lysozyme, and lay a foundation for microbial fermentation to produce lysozyme and realize industrial production.

## 2. Materials and Methods

### 2.1. Materials and Reagents

Feed liquid: fermentation broth liquid of recombinant *Pichia pastoris* strain ncy-2. Resin: D152 ion exchange resin, Beijing Solabao Technology Co., Ltd., Beijing, China.

Micrococcus dissolving wall: Institute of Microbiology, Chinese Academy of Sciences. Lysozyme Detection Kit: Nanjing Jiancheng Bioengineering Institute, Nanjing, China.

*2.2. Instruments and Equipment*

Biofilm: 30 kDapolyethersulfone(PES)polysulfone ultrafiltration membrane, Shanghai Mosu Scientific Equipment Co., Ltd., Shanghai, China. Membrane separation equipment: ro-nf-vf-40, Shanghai Mosu Scientific Equipment Co., Ltd., Shanghai, China. Ultraviolet visible spectrophotometer: uv-5100, Shanghai Yuanxie Instrument Co., Ltd., Shanghai, China. Mechanical stirring bioreactor: 10jsa, Shanghai Baoxing Biological Equipment Engineering Co., Ltd., Shanghai, China. High-speed refrigerated centrifuge: 20 pr-52 d, Hitachi Co., Ltd., Tokyo, Japan. Gradient mixer: th-500, Shanghai Huxi Analytical Instrument Factory Co., Ltd., Shanghai, China. CNC drip automatic part collector: sbs-100, Shanghai Huxi Analytical Instrument Factory Co., Ltd., Shanghai, China. Timing digital display constant flow pump: hl-2d, Shanghai Huxi Analytical Instrument Factory Co., Ltd., Shanghai, China.

*2.3. Methods*

2.3.1. Preparation of Feed Solution

A recombinant Pichia pastoris strain ncy-2 producing lysozyme was used for continuous fermentation in a 10 L bioreactor for 120 h to obtain yeast fermentation broth. The solid–liquid separation was carried out at 5000 r/min and 4 °C to remove yeast cells and various insoluble impurities in the fermentation broth. The removal rate of the cells was calculated using the blood cell counting plate method.

2.3.2. Preliminary Separation

In addition to secreting the target product, strain ncy-2 could also secrete other extracellular proteases. The purpose of this study was to isolate and extract lysozyme from the fermentation broth. Considering that the feed solution contains a large number of highly viscous substances such as proteins and sugars, in order to ensure the membrane flux of the biofilm, the PESmembrane with a molecular weight of 30 kDa was selected, and macromolecular substances such as protease greater than 30 kDa were preliminarily removed. The enzyme activities of the intercepted solution and permeate were measured to evaluate the separation effect of the biofilm. The supernatant was ultrafilteredusing the operation equipment shown in Figure 1 [9–11].

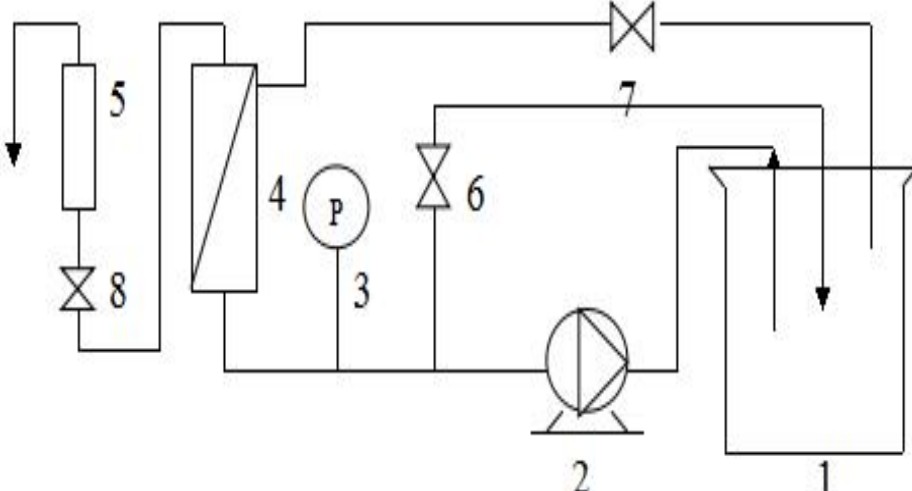

**Figure 1.** Schematic representation of the ultrafiltration equipment: 1—feed tank; 2—pressure pump; 3—pressure gauge; 4—membrane module; 5—flowmeter; 6—circulating valve; 7—concentration valve; 8—outflow valve; P—Pressure gauge.

Effect of Transmembrane Pressure on Membrane Separation Effect

The pH was fixed at 6.5, and pressure P was changed to 0.10, 0.20, and 0.30 MPa to explore the change of biofilm flux with time and its relationship with lysozyme content. Enzyme activity, membrane flux, rejection rate, transmittance, and yield were also calculated.

Effect of pH on Membrane Separation

Pressure P was set as 0.20 MPa, and pH was varied to 4.5, 6.5, and 11.0 to explore the change in biofilm flux with time and its relationship with lysozyme content. Enzyme activity, membrane flux, rejection rate, transmittance, and yield were also calculated.

### 2.3.3. Ion Exchange Chromatography

The filtrate obtained from the biofilm was purified using ion exchange chromatography. The purpose of separating and extracting lysozyme was realized using a weak acid group combined with lysozyme and an ion exchange medium. D152 resin was selected to optimize the single factor, design the response orthogonal experiment, construct the model, analyze and verify, and optimize the process parameters.

Single-Factor Experiment

Single-factor experiments were carried out with the ratio of resin dosage to liquid volume, stirring speed, pH, temperature, treatment time, and NaCl concentration as the investigation factors [12,13].

1. Ratio of resin dosage to liquid volume

The amounts of resin, fixed speed, temperature, and other conditions for ion exchange chromatography were varied, and the content of lysozyme in the eluent was determined. The recovery rate of enzyme activity was calculated, and the influence of the ratio of resin amount to solid–liquid volume on the extraction was explored.

2. Stirring speed

The stirring speed, fixed resin dosage, temperature, and other conditions for ion exchange chromatography were used to determine the lysozyme content in the eluent, calculate the enzyme activity recovery, and explore the influence of stirring speed on the extraction.

3. pH

The pH value was changed; the stirring speed, temperature, and other conditions were fixed for ion exchange chromatography; the lysozyme content in the eluent was determined; the recovery of enzyme activity was calculated; and the effect of pH on the extraction was explored.

4. Temperature

The temperature, fixed resin dosage, stirring speed, and other conditions for ion exchange chromatography were varied, and the content of lysozyme in the eluent was determined. The enzyme activity recovery was calculated, and the influence of temperature on the extraction was explored.

5. Processing time

Ion exchange chromatography was carried out under the conditions of fixed resin dosage, stirring speed, and temperature. Samples were procured every 20 min to determine the change in lysozyme content, calculate the recovery rate of enzyme activity, and explore the influence of treatment time on the extraction.

6. NaCl concentration

Ion exchange chromatography was carried out under the conditions of fixed resin dosage, stirring speed, and temperature. The concentration of the NaCl solution was varied

for elution. The change in lysozyme content in the eluent was measured, the recovery of enzyme activity was calculated, and the effect of eluent concentration on the lysozyme extraction was explored.

Orthogonal Experiment

Based on the single-factor experiment and referring to the response surface optimization method [14], a three-factor and three-level response surface analysis experiment [15–17] was designed with the enzyme activity recovery of lysozyme as the response value, and a (resin dosage in volume ratio of feed solution) (%), B (adsorption time) (h), and C (NaCl concentration) (mol/L) as variables, as shown in Table 1.

**Table 1.** Level of response surface experimental factors.

| Level | Factor | | |
| --- | --- | --- | --- |
| | A (Ratio of Resin Dosage to Liquid Volume) in % | B (Adsorption Time) in h | C (NaCl Concentration) in mol/L |
| −1 | 15 | 4 | 0.5 |
| 0 | 20 | 6 | 1.0 |
| 1 | 25 | 8 | 1.5 |

2.3.4. Preparation of Dry Enzyme Powder

Using the eluent obtained via ion exchange chromatography as a raw material, lysozyme concentrate was obtained by treatment with a polysulfone membrane, with a molecular weight of 5 kDa, and vacuum drying at −30 °C to prepare lysozyme dry enzyme powder.

Enzymatic Properties

Using dry enzyme powder as the research object, the changes in lysozyme activity under different temperatures, pH, metal ions, and surfactants were measured [18–20].

1. Effect of temperature on enzyme activity and thermal stability

Lysozyme solution (pH 6.2) was prepared with 0.01 mol/L phosphate buffer, the temperature was changed, samples were procured every 30 min, the lysozyme activity was measured, and the time change curve of lysozyme at different temperatures was drawn.

2. Effect of pH on enzyme activity and pH stability

The pH value of the phosphate buffer was adjusted, the lysozyme activity was determined, and the relationship curve between the pH value and lysozyme activity was drawn.

3. Effect of metal ions on enzyme activity

$FeSO_4$, NaCl, KCl, $CaCl_2$, $MgSO_4$, $CuSO_4$, $ZnSO_4$, $MnSO_4$, and $FeCl_2$ at a concentration of 0.01 mol/L each were used to prepare a lysozyme solution such that the final concentration of each metal ion was 5 mmol/L. Lysozyme activity was determined, and the effects of different metal ions on lysozyme activity were explored.

4. Effect of surfactants on enzyme activity

Glycerol, Tween 20, Tween 80, and Span 80 were added to the lysozyme solution such that the final concentration of each surfactant was 0.5 mg/mL. Lysozyme activity was measured to explore the effects of different surfactants on lysozyme activity.

Bacteriostatic Test

Lysozyme microspheres were activated to ensure that the strain recovered its original activity. Using lysozyme dry enzyme powder as a sample, the Oxford cup bacteriostatic experiment was carried out, the size of the bacteriostatic circle was measured, and the bacteriostatic effect of lysozyme and natural lysozyme on lysozyme microspheres was explored.

*2.4. Analysis Method*

2.4.1. Calculation of Enzyme Activity

The lysozyme content was determined according to the kit operation manual [21], that is, the enzyme activity.

$$U = \frac{UT_{15} - OT_{15}}{ST_{15} - OT_{15}} \times 200 U \cdot mL^{-1} \times N,$$

where U is in u/mL, $UT_{15}/ST_{15}/OT_{15}$ is the transmittance of the liquid to be tested/standard/blank; 200 u/mL is the standard concentration (2.5 μg/mL); and N is the dilution multiple of the solution to be tested.

$$U = \frac{U_1}{U_0} \times 100\%,$$

where U is in %; $U_1$ is enzyme activity after treatment in u/mg; and $U_0$ is primary enzyme activity in u/mg.

$$U = \frac{m_1 - m_2}{V},$$

where U is the enzyme content (mg/mL), $m_1$ is the mass of the sample before vacuum drying (g), $m_2$ is the mass of dry mass and constant weight of the sample (g), and V is the total sample volume (mL).

2.4.2. Evaluation Parameters of Membrane Performance

Membrane flux [22] is the volume of permeate per unit membrane area per unit time. The experimental operation parameters were changed, samples were taken after stable operation, and the volume of the permeate was recorded within a certain time. The formula used for calculation is:

$$J_w = \frac{V}{S_m t},$$

where $J_w$ is the membrane flux, L/(m²·h); V is the total volume of permeate, L; $S_m$ is the effective area of the membrane, m²; and t is the filtering time, h.

Retention rate (the retention capacity of membrane to solute) is expressed as decimal or percentage. Concentration polarization exists in the actual membrane separation process, and the real rejection rate is:

$$R_0 = 1 - \frac{C_p}{C_m},$$

Because it is difficult to determine the polarization concentration $C_m$, the volume concentration of the feed solution was used to replace the polarization concentration, and the apparent rejection rate R was used to replace the rejection rate $R_0$.

$$R = 1 - \frac{C_p}{C_b},$$

where $R_0$ is the rejection rate, R is the apparent rejection rate, and $C_p$, $C_b$, and cm are the permeate, feed solution, and membrane surface concentrations, respectively (mol/L).

Concentration multiple CF and yield REC [23] are determined as follows:

$$CF = \left(\frac{C_0}{C}\right)^R,$$

$$REC = \left(\frac{V_0}{V}\right)^{R-1},$$

where $C_0$ and C are the concentration of the feed solution and concentrated solution, respectively (mol/L), and $V_0$ and V are the volume of the feed solution and concentrated solution, respectively.

## 3. Results and Discussion

### 3.1. Preparation of Feed Solution

The cell concentration in the fermentation broth was $4.675 \times 10^9$ cells/mL, and after solid–liquid separation, it decreased to $8.25 \times 10^5$ cells/mL, and the removal rate of bacterial cells reached 99.98%. All kinds of insoluble impurities such as bacterial cells and fermentation residues in the fermentation broth were preliminarily removed.

### 3.2. Preliminary Separation

3.2.1. Effect of Transmembrane Pressure on Membrane Separation Effect

Effect of Different Pressures on the Extraction of Microbial Enzymes

As shown in Figure 2, the transmembrane pressure had a significant impact on the membrane flux and lysozyme activity. When the transmembrane pressure was 0.20 MPa, the maximum membrane flux was 38.2 $L \cdot (M^2 \cdot h)^{-1}$, and the enzyme activity was the highest. In the early stage, owing to the increase in transmembrane pressure, the shear force on the membrane surface increased; the feed liquid velocity accelerated, showing a turbulent state;the concentration polarization phenomenon on the membrane surface decreased, resulting in an increase in membrane flux; and agel layer was formed on the surface of the membrane. The higher the pressure, the higher the density and thickness of the gel layer, and the worse the polarization phenomenon; the larger the resistance, the lower the flow rate of the liquid material and the longer the time, and the lower the membrane flux, resulting in aloss of and decrease in enzyme activity. However, excessive pressure leads to serious membrane pollution, difficult membrane cleaning, and increased energy consumption, and it affects the activity of lysozyme. Therefore, it is very important to select the appropriate pressure.

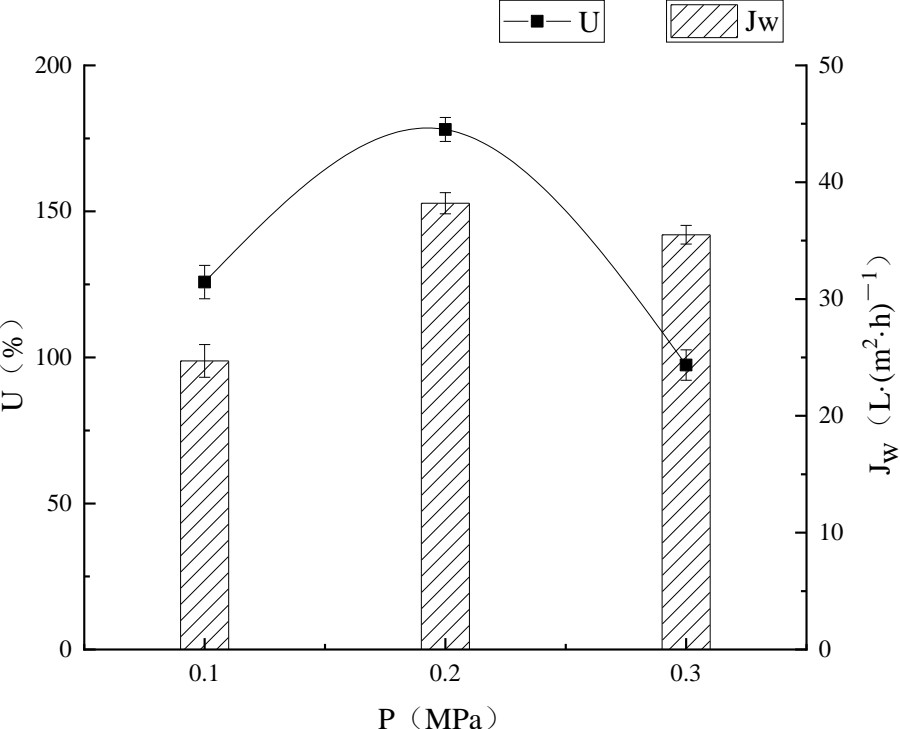

**Figure 2.** Effect of transmembrane pressure on ultrafiltration separation effect.

Effect of Different Pressures on Membrane Flux

As shown in Figure 3, the membrane flux increased with an increase in pressure. When the pressure rose to 0.30 MPa, the membrane flux was lower than 0.20 MPa, which was due to the concentration polarization on the membrane surface. Under the same pressure, the flux trend of the membrane increased rapidly and then decreased slowly. The higher the

initial pressure, the higher the membrane flux. The gel layer formed at a later stage of the membrane. As the resistance of the ultrafiltration membrane increased, the concentration polarization phenomenon on the surface of the membrane increased, and the flux of the membrane decreased.

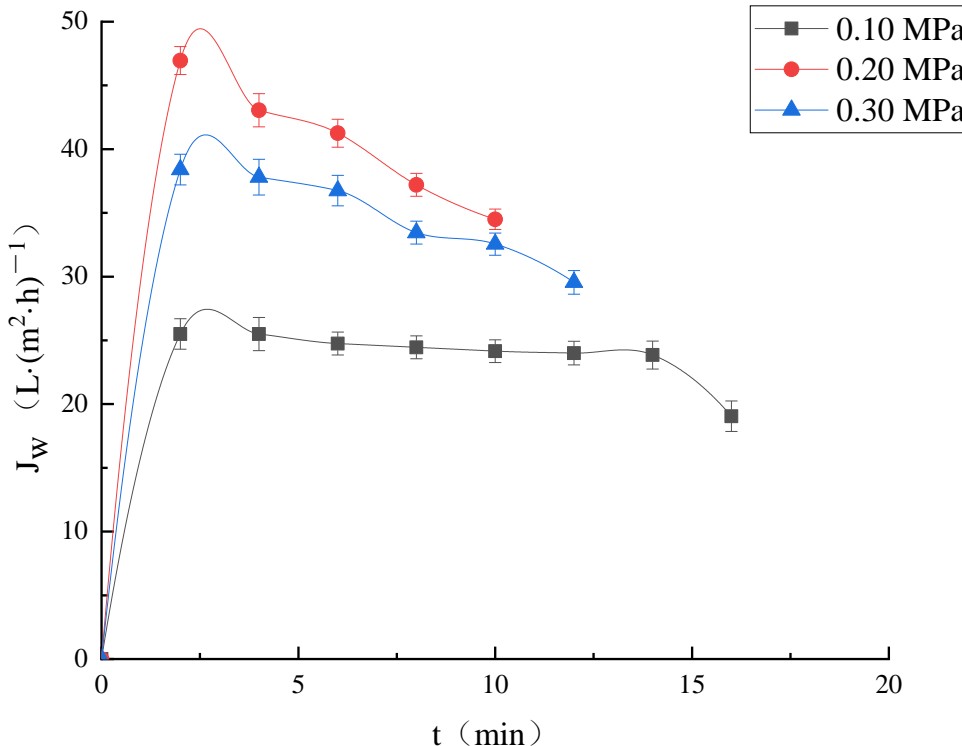

**Figure 3.** Variation curve of membrane flux with time under different pressures.

Effect of Different Pressures on Ultrafiltration

As shown in Table 2, different operating pressures had different effects on the yield of lysozyme, especially the rejection and transmittance. Owing to the difference in transmembrane pressure, the thickness and density of the gel layer were different, and the resistance was also different, thus affecting the rejection rate and transmittance. At the pressure of 0.20 MPa, the minimum interception rate of lysozyme was 7.4%. The maximum transmittance was 45.7% and the yield was 96.6%.

**Table 2.** Effect of transmembrane pressure on ultrafiltration effect.

| P (MPa) | R (%) | R′ (%) | REC (%) |
|---------|-------|--------|---------|
| 0.10 | 11.2 | 37.4 | 93.6 |
| 0.20 | 7.4 | 45.7 | 96.6 |
| 0.30 | 15.85 | 37.7 | 91.9 |

P—pressure R—rejection rate; R′—transmittance; REC—yield.

### 3.2.2. Effect of pH on Membrane Separation Effect

Effect of Different pH Values on the Extraction of Lysozyme

As shown in Figure 4, the pH of the feed solution had a significant influence on the membrane separation. When the membrane flux and lysozyme activity in the permeate at pH 4.5 and 11.0 were lower than that at pH 6.5, the maximum membrane flux was 38.2 L·$(M^2 \cdot h)^{-1}$, and the lysozyme activity was the highest. Under the condition of pH 4.5, almost no charge, aggregation occurred easily, forming a gel layer on the surface of the membrane and increasing resistance, resulting in a decrease in membrane flux. When the pH value was 6.5, it was negatively charged, and there was an electrostatic repulsion between the molecules. Concurrently, other hetero-proteins were also be negatively

charged, which weakened the concentration polarization phenomenon to a certain extent and ensured high membrane flux. In addition, the isoelectric point of lysozyme was 11, without charge, and the repulsive force between molecules was the smallest. It was easy to form a gel layer by molecular flocculation, and the concentration polarization phenomenon was aggravated, and the membrane flux decreased.

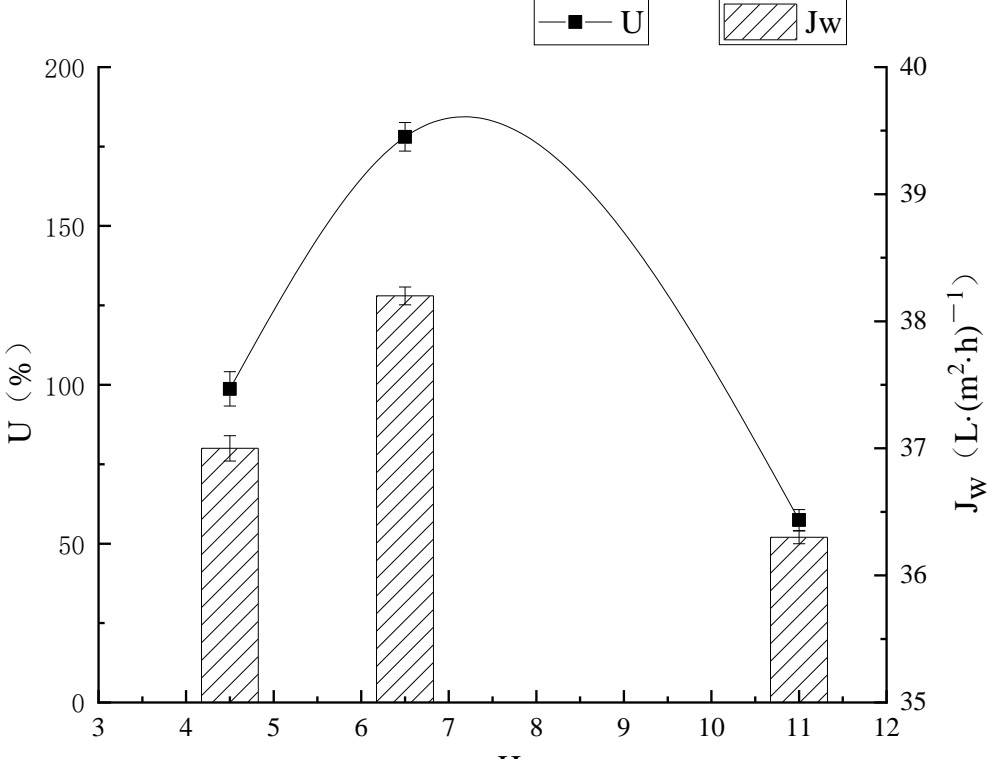

**Figure 4.** Effect of pH on ultrafiltration separation.

Effect of Different pH on Ultrafiltration Membrane Flux

As shown in Figure 5, the membrane flux decreased with time. At pH 6.5, lysozyme was negatively charged and was relatively stable between molecules, ensuring a high membrane flux. At pH 4.5, molecular aggregation easily occurred and was attached to the membrane surface, resulting in a decrease in the membrane flux. The charge of lysozyme at the isoelectric point was 0, the repulsion between molecules was the smallest, intermolecular flocculation occurred easily, and the membrane flux was reduced.

Effect of Different pH Values on Ultrafiltration

As shown in Table 3, pH had a significant influence on the interception rate of lysozyme. When the pH value was 6.5, the rejection and transmittances were 7.4% and 45.7%, respectively, and the yield of lysozyme was 96.6%. This was due to the negative charge of lysozyme and the electrostatic interaction between enzyme molecules, resulting in concentration polarization on the membrane surface, which affected the membrane separation effect.

**Table 3.** Effect of pH on ultrafiltration.

| pH | R (%) | R′ (%) | REC (%) |
|----|-------|--------|---------|
| 4.5 | 16.2 | 48.5 | 92.1 |
| 6.5 | 7.4 | 45.7 | 96.6 |
| 11 | 15.3 | 47.7 | 93 |

R—rejection rate; R′—transmittance; REC—yield.

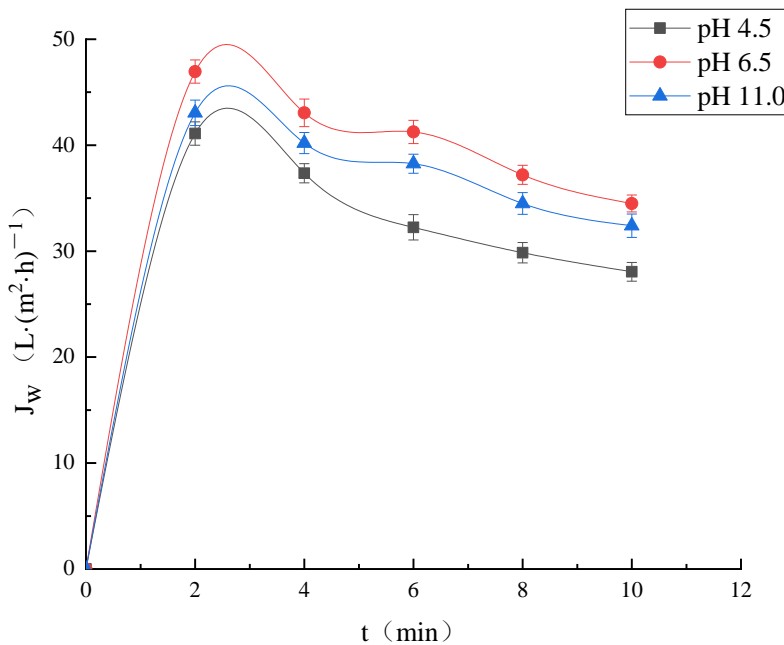

**Figure 5.** Variation curve of membrane flux with time at different pH values.

### 3.3. Ion Exchange Chromatography

3.3.1. Single Factor Experiment

Ratio of Resin Dosage to Liquid Volume

As shown in Figure 6, the ability of the resin to adsorb and desorb enzyme molecules first increased and then decreased. In the adsorption stage, when the amount of resin reached 20%, the growth of resin adsorption capacity slowed down; when it exceeded 25%, it did not increase, but decreased, indicating that the resin had been adsorbed and was saturated. In the elution stage, when the amount of resin reached 20%, the elution was complete.

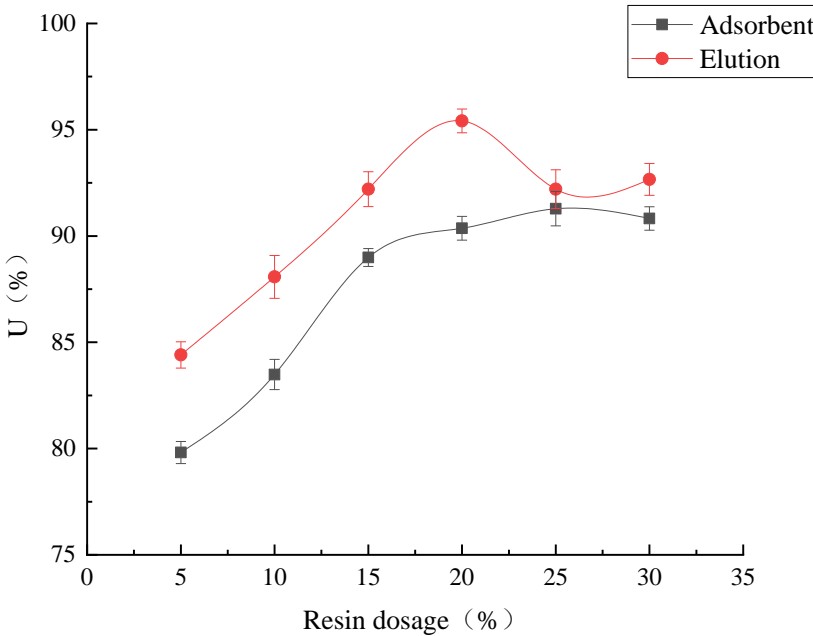

**Figure 6.** Effect of the ratio of resin dosage to liquid volume on the extraction of lysozyme.

Stirring Speed

As can be seen from Figure 7, the external diffusion of lysozyme increased with an increase in the stirring speed of the feed solution, the resin exchange capacity increased, and the internal diffusion was not affected.

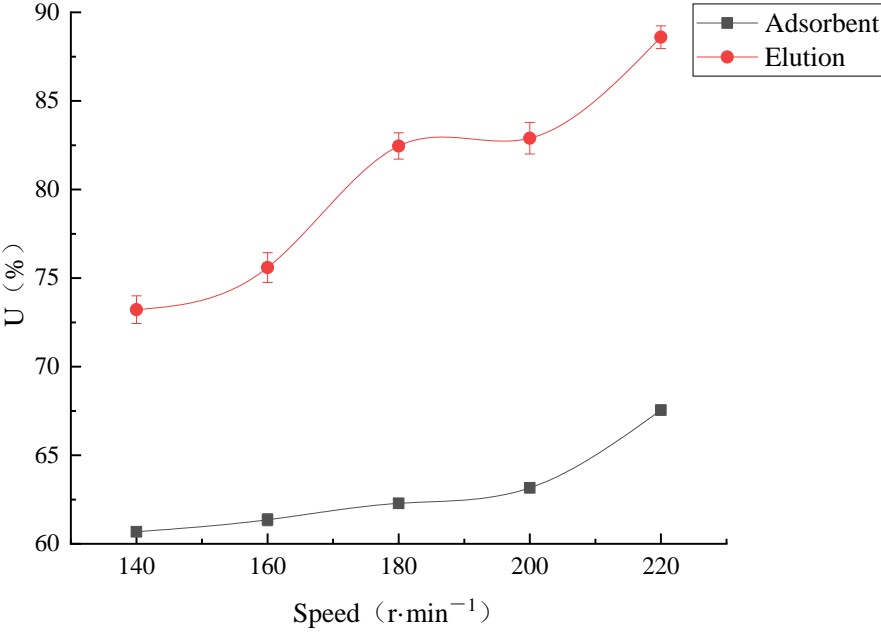

**Figure 7.** Effect of stirring speed on extraction of lysozyme.

pH

The pH of the feed solution determined the dissociation strength between the resin active group and the exchange ion and affected the exchange capacity of the resin. When the pH changed, the intermolecular forces changed, and the charges differed. As shown in Figure 8, with an increase in the pH of the feed solution, the swelling degree with a high degree of hydration was large, and the resin exchange capacity increased first and then decreased.

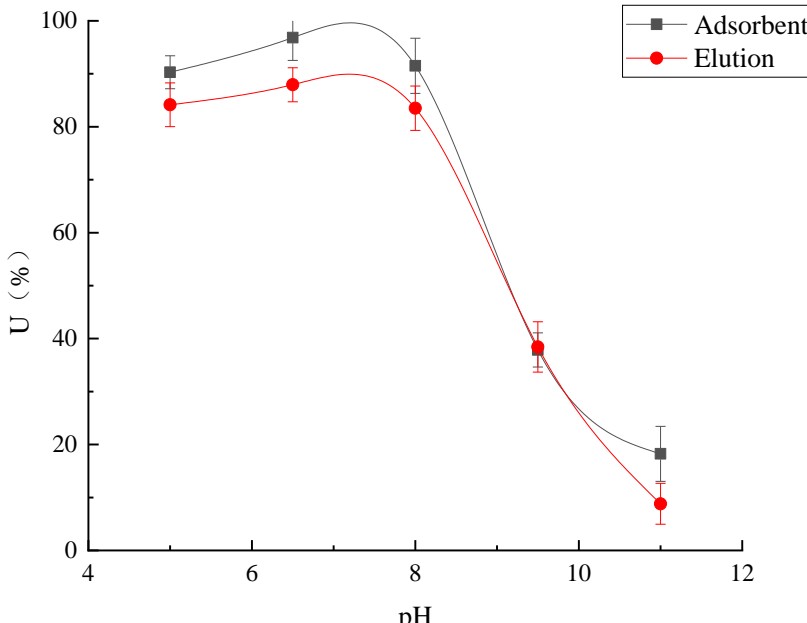

**Figure 8.** Effect of pH on extraction of lysozyme.

Temperature

As shown in Figure 9, the adsorption saturation of the resin was sensitive to temperature. With an increase in the feed liquid temperature, the diffusion and exchange speed also accelerated, and the enzyme activity decreased when the time was too long at too high a temperature.

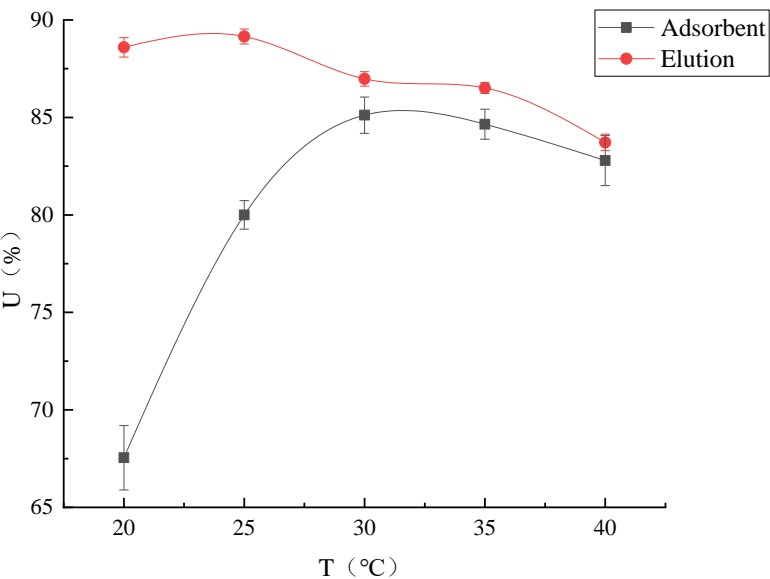

**Figure 9.** Effect of temperature on extraction of lysozyme.

Processing Time

As shown in Figure 10, after the resin reached saturation, the enzyme activity remained unchanged or even decreased, which was caused by long-time adsorption, partial inactivation of enzymes, or human factors. At the initial stage of analysis, owing to the large concentration difference between the resin and the lysozyme in the eluent, the desorption power was large, and the speed was fast, but the enzyme activity decreased after 80 min of elution and increased after 100 min of elution, which was due to the uneven distribution caused by salting out, resulting in an increase in enzyme activity.

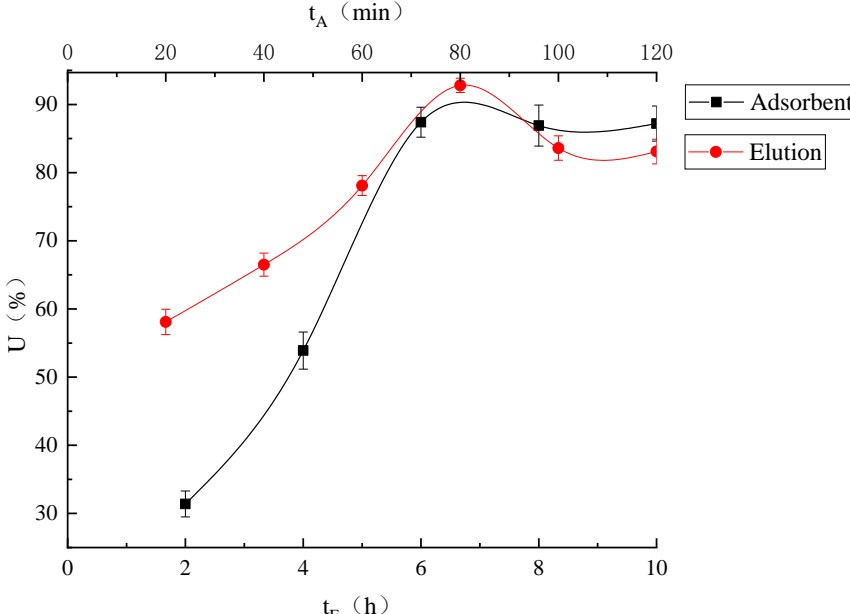

**Figure 10.** Effect of treatment time on extraction of lysozyme.

NaCl Concentration

As shown in Figure 11, the degree of enzyme desorption was significantly affected by the NaCl concentration. When the NaCl concentration reached 1.0 mol/L, the enzyme activity recovery, reaching the maximum value, and the concentration continued to increase but then decreased. This is because the salt concentration in the feed solution was too high, resulting in the aggregation of lysozyme molecules and affecting desorption.

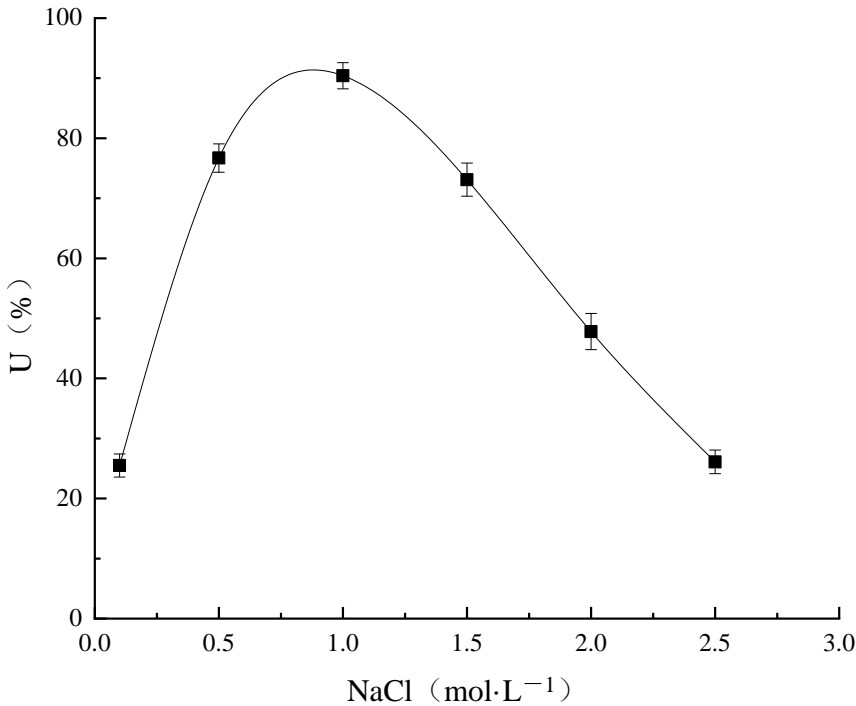

**Figure 11.** Effect of NaCl concentration on extraction of lysozyme.

3.3.2. Orthogonal Experiment
Establishment of Regression Model

A response surface optimization method was used to design the experimental scheme. The specific results are presented in Tables 4 and 5.

**Table 4.** Response surface experimental design scheme and results.

| Number | A | B | C | U/% |
|---|---|---|---|---|
| 1 | 0 | 1 | −1 | 43.432 |
| 2 | −1 | 0 | −1 | 66.8512 |
| 3 | 1 | 1 | 0 | 71.8863 |
| 4 | −1 | −1 | 0 | 94.5497 |
| 5 | 0 | −1 | 1 | 77.8369 |
| 6 | 0 | 0 | 0 | 92.4846 |
| 7 | 0 | 0 | 0 | 81.9566 |
| 8 | 0 | 1 | 1 | 67.7667 |
| 9 | 1 | 0 | 1 | 78.9759 |
| 10 | −1 | 0 | 1 | 71.6521 |
| 11 | 0 | 0 | 0 | 86.7681 |
| 12 | −1 | 1 | 0 | 76.006 |
| 13 | 0 | −1 | −1 | 62.9551 |
| 14 | 1 | 0 | −1 | 49.2229 |
| 15 | 1 | −1 | 0 | 89.2804 |
| 16 | 0 | 0 | 0 | 85.6185 |
| 17 | 0 | 0 | 0 | 85.1607 |

**Table 5.** Variance analysis of quadratic response surface regression model.

| Source | SS | Df | MS | F | *p* |
|---|---|---|---|---|---|
| Model | 3195.16 | 9 | 355.02 | 37.15 | <0.0001 |
| A | 48.48 | 1 | 48.48 | 5.07 | 0.059 |
| B | 536.79 | 1 | 536.79 | 56.17 | 0.0001 |
| C | 680.26 | 1 | 680.26 | 71.18 | <0.0001 |
| AB | 0.33 | 1 | 0.33 | 0.035 | 0.8578 |
| AC | 155.65 | 1 | 155.65 | 16.29 | 0.005 |
| BC | 22.34 | 1 | 22.34 | 2.34 | 0.1702 |
| $A^2$ | 0.047 | 1 | 0.047 | $4.893 \times 10^{-3}$ | 0.9462 |
| $B^2$ | 53.74 | 1 | 53.74 | 5.62 | 0.0495 |
| $C^2$ | 1655.29 | 1 | 1655.29 | 173.19 | <0.0001 |
| Residua | 66.9 | 7 | 9.56 | | |
| Lack of Fit | 7.85 | 3 | 2.62 | 0.18 | 0.9065 |
| Pure Error | 59.05 | 4 | 14.76 | | |
| Cor Total | 3262.06 | 16 | | | |
| $R^2$ | 0.9795 | | | | |
| $R^2$ (adj) | 0.9531 | | | | |
| C.V.% | 0.041 | | | | |

A quadratic response surface regression analysis was carried out on the data in Table 4 using Design-Expert software, and amultivariate quadratic regression model of lysozyme activity Y on resin dosage in feed liquid volume ratio a, adsorption time B, and NaCl concentration C was obtained: $Y = 60.132 - 3.32861A + 3.68373B + 112.97956C + 0.028742AB + 2.49521AC + 2.36321BC + 4.21546 \times 10^{-3}A^2 - 0.89312B^2 - 79.31020C^2$ ($p< 0.0001$, $R^2 = 0.9795$, $R^2$ (adj) = 0.9531 (>0.80), CV = 4.10%.

As shown in Table 5, according to the equation model results, the model significance test value f was 37.15, the significance level $p< 0.0001$, and the model term $p \leq 0.05$, indicating that the established model regression equation was significant and statistically significant; the mismatch term $p > 0.05$, the correlation coefficient $R^2 = 0.9795$, the adjusted correlation coefficient $R^2$ (adj) = 0.9531 (>0.80), and coefficient of variation CV = 4.10%, indicating that the model had good fit, the proportion of abnormal error between the model and the actual fitting was small, and the mismatch term was not significant. The model equation can be used to preliminarily analyze and predict the process of lysozyme extraction.

Regression Model Analysis

We fixed any factor and analyzed the deformation of the regression equation. The results are as follows.

It can be seen from the response surface and contour map in Table 5 and Figures 12–14 that there was no significant difference in the interaction between the resin dosage volume ratio a and the adsorption time B, the interaction between the resin dosage volume ratio a and the NaCl concentration C was extremely significant, and the interaction between the adsorption time B and the NaCl concentration C was not significant. It can be seen from the contour line that the extraction rate y was less sensitive to the change of adsorption time B than to the change of the ratio of resin dosage to feed liquid volume a, and it was more sensitive to the change of NaCl concentration C than to the change of the ratio of resin dosage to feed liquid volume a. The interaction between adsorption time B and NaCl concentration C was more sensitive than that of the other two groups. Therefore, the sequence of three factors of resin dosage in volume ratio of feed solution a, adsorption time B, and NaCl concentration C affecting the enzyme activity recovery y of lysozyme was C > B> A.

Validation Experiment

The data were analyzed using Design-Expert (StatEase company, Minneapolis, MN, USA) and SPSS (International Business Machines Corporation, Armonk, New York, NY, USA) software, and the comparison results are shown in the table below.

It can be seen from Table 6 that the analysis results of the Design-Expert and SPSS were consistent. The NaCl concentration in the regression analysis of variance table was $p < 0.0001$, the main effect test of SPSS analysis was $p = 0.022$, the multiple comparison results of NaCl concentration were $p = 0.011$ (all $< 0.05$) when the concentration was 1.0 mol/L, and the most influential factor was NaCl concentration.

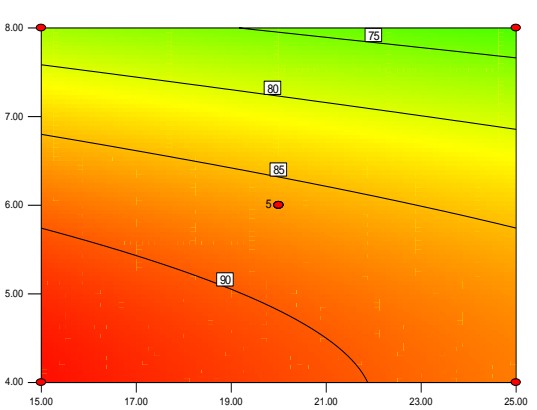
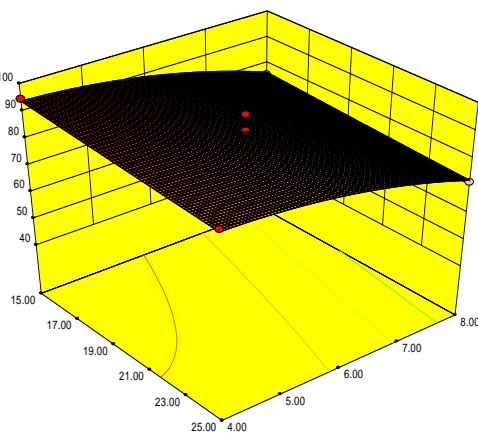

**Figure 12.** Contour and response surface of resin dosage in volume ratio of material to liquid and adsorption time. The red dots in the figure are the salient points of the contour map and response surface map.

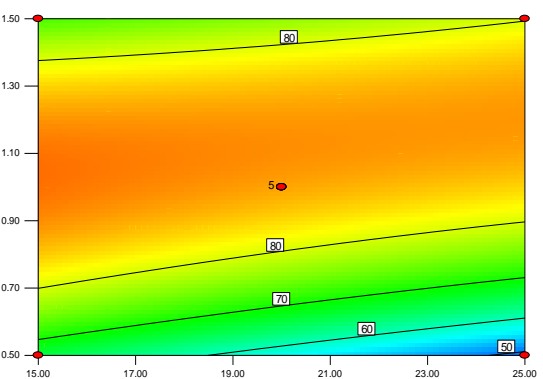
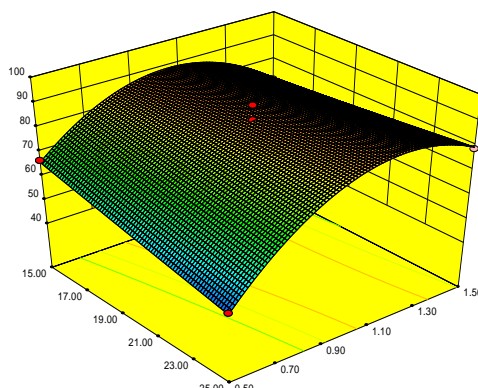

**Figure 13.** Contour and response surface of resin dosage in volume ratio of material to liquid and NaCl concentration. The red dots in the figure are the salient points of the contour map and response surface map.

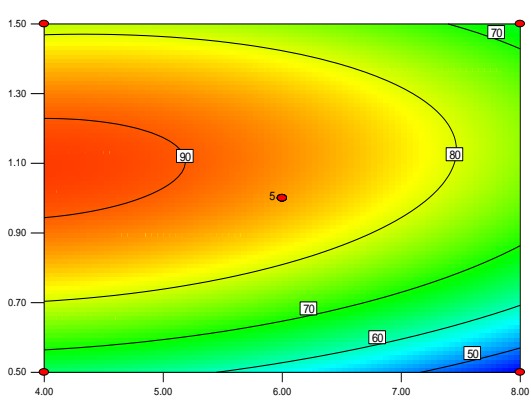
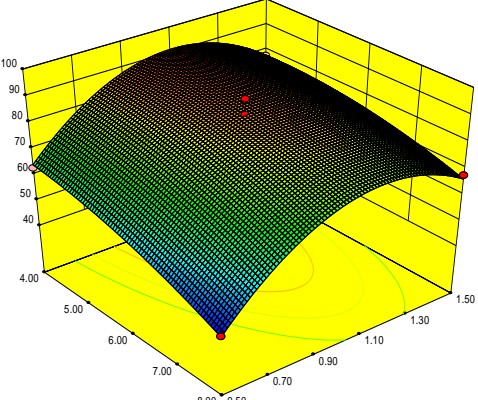

**Figure 14.** Contour and response surface of adsorption time and NaCl concentration. The red dots in the figure are the salient points of the contour map and response surface map.

**Table 6.** Comparison of Design-Expert and SPSS software results.

| Software | R² | *p* | Significant Factor |
|---|---|---|---|
| Design-Expert | 0.9795 | <0.05 | NaCl |
| SPSS | 0.9820 | <0.05 | NaCl |

By solving the regression equation, the optimum theoretical conditions were as follows: the amount of resin accounted for 15% of the volume of the feed solution, the adsorption time was 4.0 h, the concentration of NaCl was 1.01 mol/L, and the recovery of enzyme activity was 93.88%. Considering the actual operation, the amount of resin accounted for 15% of the volume of the feed solution, the adsorption time was 4 h, and the NaCl concentration was 1 mol/L. Under these conditions, the recovery of enzyme activity was 95.67%, and the error was 0.22% (<1%). The process conditions established in this experiment are reliable, and the model fits well with the actual situation.

### 3.4. Preparation of Dry Enzyme Powder

The prepared lysozyme dry enzyme powder was milky white, and the lysozyme content was 33.8 mg/mL. The enzyme solution was prepared with 0.005 g/mL. The enzyme activity was 12,573.6 u/mg.

3.4.1. Enzymatic Property Test

Effect of Temperature on Enzyme Activity and Thermal Stability

As shown in Figure 15, in the range of 20–50 °C, the activity of lysozyme increased with an increase in temperature, which was relatively stable, and the enzyme activity remained above 50%. The temperature continued to rise, and lysozyme activity decreased. This was due to the denaturation and inactivation caused by the increase in temperature.

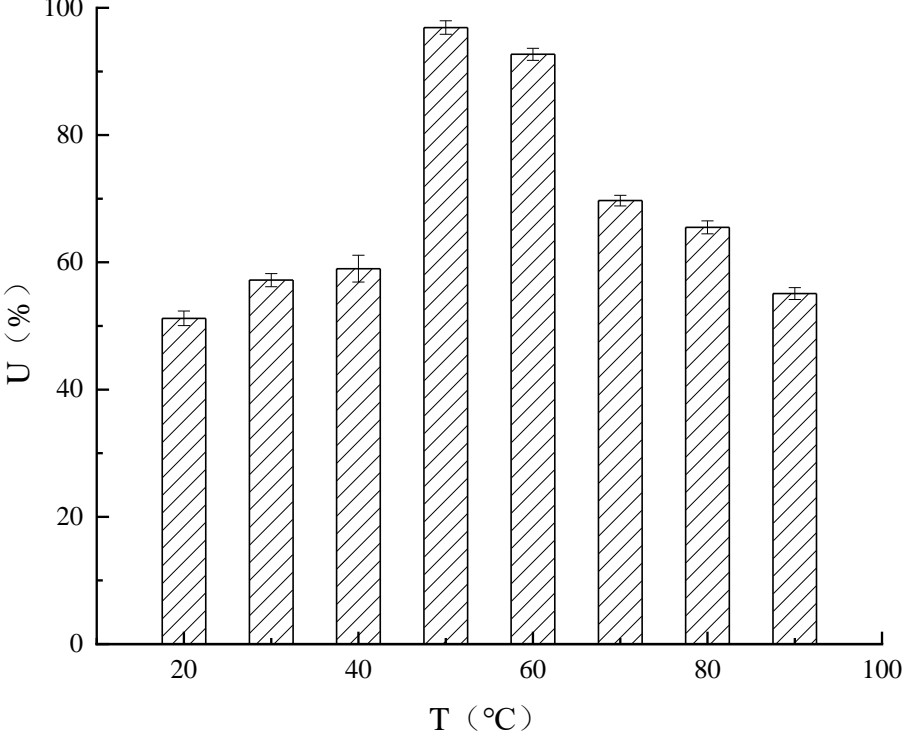

**Figure 15.** Effect of temperature on enzyme activity.

As shown in Figure 16, temperature had little effect on lysozyme activity in the range of 20–60 °C. Lysozyme had good thermal stability. After holding at 70 °C for 90 min, the enzyme activity decreased to approximately 75%, and the enzyme activity decreased continuously. At 90 °C, lysozyme was extremely unstable, denatured, and inactivated in a

short time, and the enzyme activity decreased sharply. When the temperature was lower than 60 °C, lysozyme couldstably maintain its activity.

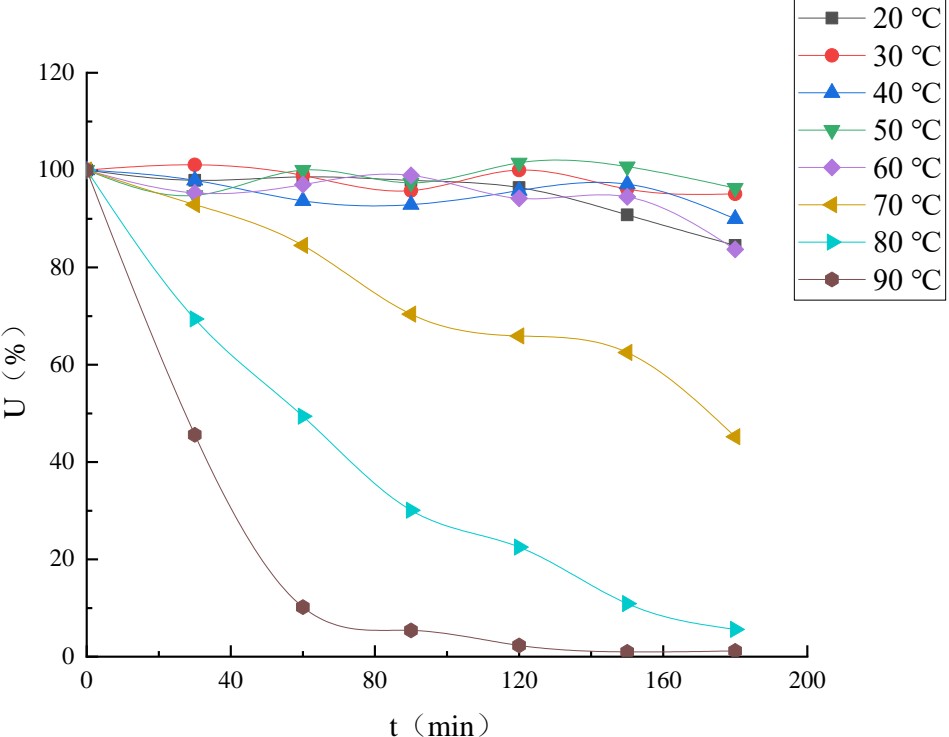

**Figure 16.** Changes in enzyme activity with time at different temperatures.

Effect of pH on Enzyme Activity and pH Stability

As shown in Figure 17, the activity of lysozyme showed a "peak" shape, which first increased and then decreased with an increase in pH. Under neutral conditions, the activity of lysozyme was the largest and most stable.

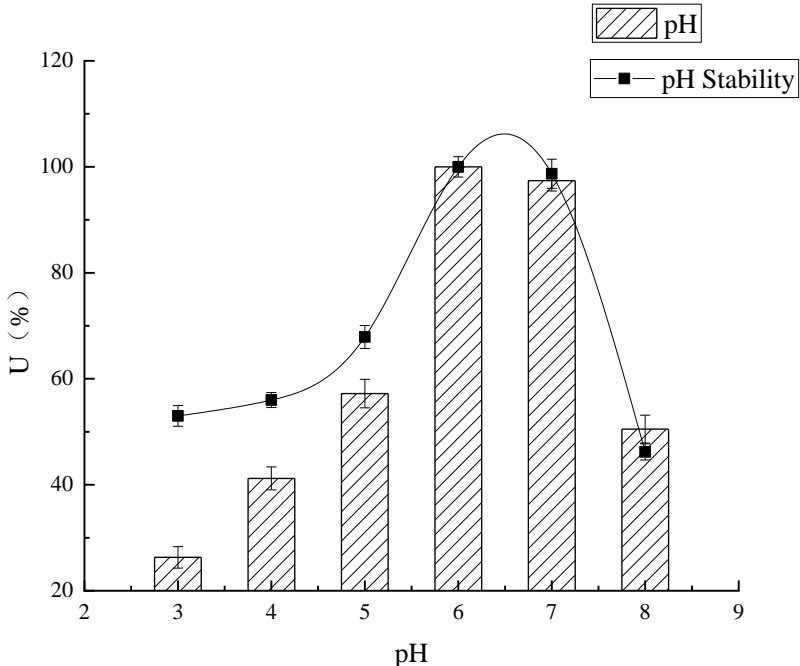

**Figure 17.** Effect of pH on enzyme activity.

Effect of Metal Ions on Enzyme Activity

As shown in Figure 18, the enzyme activity was inhibited by $Fe^{2+}$, $Fe^{3+}$, $Zn^{2+}$, and $Cu^{2+}$ and activated by $Na^+$ and $Mg^{2+}$, whereas $Mn^{2+}$, $K^+$, and $Ca^{2+}$ had no significant effect on lysozyme activity. Except for a few metal ions with strong inhibitory effects, most metal ions could pair well with lysozyme and had little effect on enzyme activity.

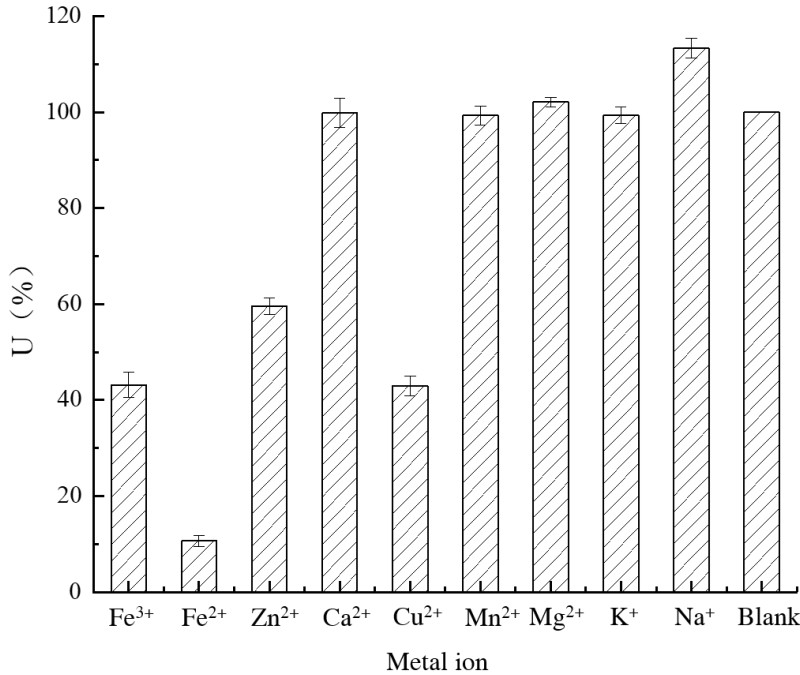

**Figure 18.** Effect of metal ions on enzyme activity.

Effect of Surfactants on Enzyme Activity

As shown in Figure 19, the overall effects of the four surfactants were similar. Span80 inhibited lysozyme activity, whereas Tween20 and 80 and glycerol activated it.

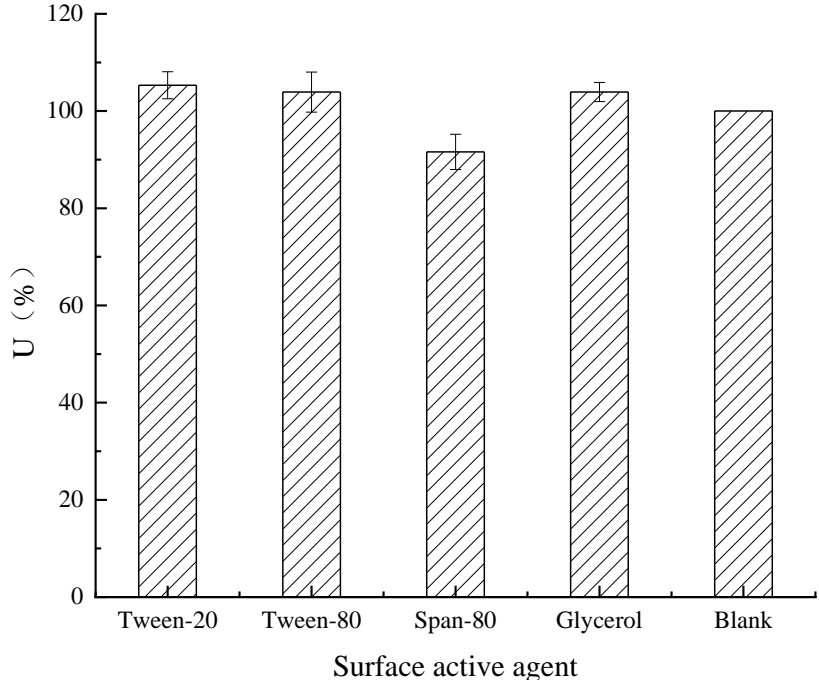

**Figure 19.** Effect of surfactant on enzyme activity.

### 3.4.2. Bacteriostatic Test

As shown in Figure 20, lysozyme dry enzyme powder prepared from the fermentation broth of the recombinant yeast strain had the same inhibitory effect on microsphere lysozyme as natural lysozyme.

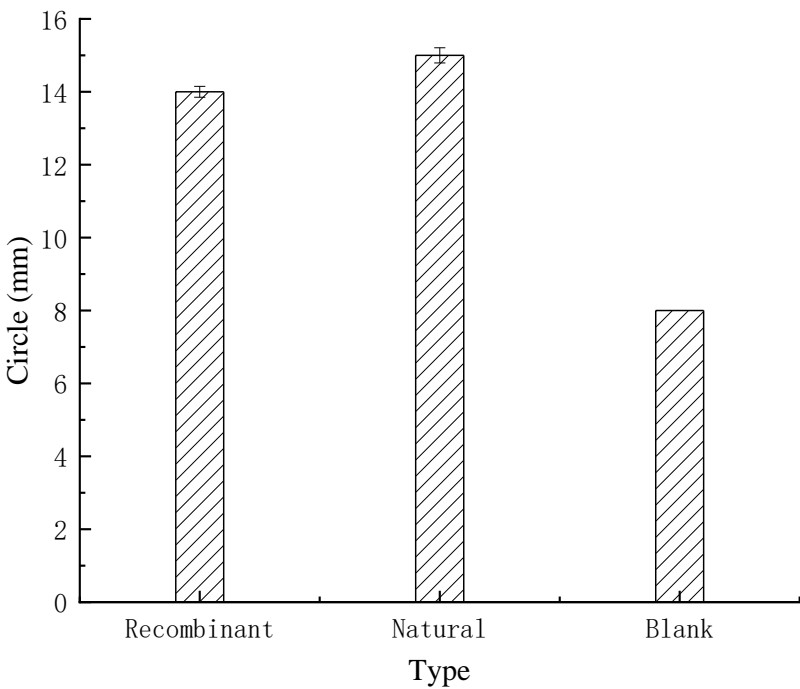

**Figure 20.** Comparison of bacteriostatic effect.

### 4. Conclusions

After solid–liquid separation, the removal rate of bacterial cells reached 99.98%. The optimal conditions of biofilm are as follows: transmembrane pressure of 0.20 MPa and feed solution pH of 6.5. The yield of lysozyme was 96.6%, and the enzyme activity was 2612.1 u/mg, which was 1.78 times that of the original enzyme. A regression equation model was established. The optimal process conditions obtained using the model were as follows: The amount of resin accounted for 15% of the volume of material liquid, the adsorption time was 4 h, and the NaCl concentration was 1.0 mol/L. Under these conditions, the recovery of lysozyme was 95.67%, the enzyme activity was 3879.6 u/mL, and the purification multiple was 0.5, which was 3.1 times that of the original enzyme. Lysozyme dry enzyme powder had an inhibitory effect on *Micrococcus* lysozyme, and the enzyme activity was 12,573.6 u/mg. The optimum temperature was 50 °C, and the thermal stability was good, in the range of 20–60 °C. The optimum pH was 6.5. The enzyme activity of lysozyme was inhibited by $Fe^{2+}$, $Fe^{3+}$, $Zn^{2+}$, and $Cu^{2+}$, while $Na^+$ and $Mg^{2+}$ activated the enzyme activity, which can also be inhibited by Span 80 and activated by Tween 20, Tween 80, and glycerol. The product obtained by the above process not only has the advantages of natural lysozyme, but also overcomes the disadvantages of low activity, low yield, inconvenient preparation, and inability to be stably preserved. It has a wide application prospect by using microbial fermentation to expand production.

**Author Contributions:** Conceptualization, H.Z. and S.C.; methodology, Y.T. and Y.Z.; software, S.C.; validation, H.Z. and J.L.; formal analysis, S.C.; investigation, Y.T.; resources, J.L.; data curation, S.C.; writing—original draft preparation, H.Z. and S.C.; writing—review and editing, H.Z.; visualization, J.L.; supervision, H.Z.; project administration, L.S.; funding acquisition, H.Z. and L.S. All authors have read and agreed to the published version of the manuscript.

**Funding:** This work was supported by the National Natural Science Foundation of China (No. 32001836), the Natural Science Foundation of Shandong Province (ZR201911180224), and the Yantai Science and Technology Development Plan (SK21H266).

**Institutional Review Board Statement:** Not applicable.

**Conflicts of Interest:** The authors declare no conflict of interest.

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
