# Peer review of "Application of Ultrafiltration and Ion Exchange Separation Technology for Lysozyme Separation and Extraction"

_fermentation, doi:10.3390/fermentation8070297_

Round 1
Reviewer 1 Report
comments attached

Reviewer 2 Report
The manuscript is focused on the application of UF and ion exchange separation technology for lysozyme separation and extraction. The presenting topic is interesting, however, the manuscript must be improved before the publication in the Fermentation Journal.
The biggest disadvantage of the presented manuscript is that it is in the form of a report, not a publication. Please, see the comments below:
1. The introduction doest not provide sufficient background:
- The introduction is too short.
- The introduction does not present the discussed topic:
a) There is no basic information about the processes: ultrafiltration and ion exchange. Moreovere, there is no information about advantages and applications of these processes.
b) There is no basic information about lysozyme.
c) The introduction does not present the results obtained so far on the research topic undertaken by the Authors.
- In this study, the authors used polymeric membranes. Explain why, discuss their advantages and disadvantages.
- The novelty of the work should be presented.
2. The introduction does not include all relevant references:
- The Authors cited only 19 papers (!) of which only 4 were published in the last 3 years (!). It is necessary to make a thorough review of the literature and present the current state of the art.
-Basic information about UF, its applications and advantages is available in the recently published articles:
doi: 10.3390/membranes10110319
doi: 10.3390/membranes12050519
-Basic information about ion exchange, its applications and advantages is available in the recently published article:
doi: 10.1016/j.memsci.2022.120325
- The information about advantages and disadvantages of polymeric membranes are thoroughly discussed in a recently published work:
doi: 10.3390/membranes11010044
3. The results are not well presented (Section 3):
- The results are discussed too briefly.
- There is no scientific discussion of the results obtained.
- There is no comparison of the obtained results with the results presented in the literature.
4. English must be improved, for instance:
- line 38: Instead of "worth fermentation" - "fermentation broth"
Round 2
Reviewer 1 Report
comments attached in the manuscript

Reviewer 2 Report
Thank you very much for your answer.
The manuscript has been improved, hence, I recommend it for publication.
